# Proportionate translation of study materials and measures in a multinational global health trial: methodology development and implementation

Ashleigh Charles ,[1] Palak Korde,[2] Chris Newby ,[3] Alina Grayzman,[4] Ramona Hiltensperger ,[5] Candelaria Mahlke ,[6] Galia Moran ,[4] Juliet Nakku ,[7,8] Jackie Niwemuhwezi,[7] Rebecca Nixdorf ,[6] Eva Paul ,[5] Bernd Puschner ,[5] Mary Ramesh,[9] Grace Kathryn Ryan ,[10] Donat Shamba ,[9] Jasmine Kalha ,[2] Mike Slade [1]

For numbered affiliations see end of article.

**Correspondence to**
Ms Ashleigh Charles;
Ashleigh.charles@nottingham.ac.uk

## ABSTRACT

**Objectives** Current translation guidelines do not include sufficiently flexible translation approaches for different study materials. We aimed to develop a proportionate methodology to inform translation of all types of study materials in global health trials.

**Design** The design included three stages: (1) categorisation of study materials, (2) integration of existing translation frameworks and (3) methodology implementation (Germany, India, Israel, Tanzania and Uganda) and refinement.

**Participants** The study population comprised 27 mental health service users and 27 mental health workers who were fluent in the local language in stage 7 (pretesting), and 54 bilingual mental health service users, aged 18 years or over, and able to give consent as judged by a clinician for step 9 (psychometric evaluation).

**Setting** The study took place in preparation for the Using Peer Support in Developing Empowering Mental Health Services (UPSIDES) randomised controlled trial (ISRCTN26008944).

**Primary outcome measure** The primary outcome measure was the Social Inclusion Scale (SIS).

**Results** The typology identifies four categories of study materials: local text, study-generated text, secondary measures and primary measure. The UPSIDES Proportionate Translation Methodology comprises ten steps: preparation, forward translation, reconciliation, back translation, review, harmonisation, pretesting, finalisation, psychometric evaluation and dissemination. The translated primary outcome measure for the UPSIDES Trial (SIS) demonstrated adequate content validity (49.3 vs 48.5, p=0.08), convergent validity and internal consistency (0.73), with minimal floor/ceiling effects.

**Conclusion** This methodology can be recommended for translating, cross-culturally adapting and validating all study materials, including standardised measures, in future multisite global trials. The methodology is particularly applicable to multi-national studies involving sites with differing resource levels. The robustness of

## Strengths and limitations of this study

► This paper offers a proportionate translation methodology for researchers to use when translating and validating all study materials needed in a global health trial, which is suitable for use across different resource settings and when time or research capability are limited.

► The proportionate translation methodology supports the goal of scaling up and evaluating evidence-based interventions, and increasing access to these interventions in low-income and middle-income countries.

► The methodology was implemented by its developers, meaning that some components of the methodology may be implicit knowledge which is not sufficiently described.

► The generalisability to global health trials in areas of medicine beyond mental health is unknown.

► We were only able to demonstrate preliminary psychometric adequacy of the primary outcome Social Inclusion Scale due to the limited sample sizes for each site.

the psychometric findings is limited by the sample sizes for each site. However, making this limitation explicit is preferable to the typical practice of not reporting adequate details about measure translation and validation.

**Trail registration number** ISRCTN26008944

## INTRODUCTION

A global health goal is to scale up evidence-based interventions, especially in low-income and middle-income countries and other low-resource settings,[1] in order to maximise health equity.[2] A key contribution to this goal comes from randomised controlled trials,

which provide gold-standard evidence about intervention effectiveness and, alongside other study designs,[3] illuminate the relationship between context, implementation and outcome. The translation of measures for use in global research including health trials is an important foundation for a high-quality scientific evidence base.

Multinational cooperation in clinical trials is essential.[4] Close coordination is needed due to the number of different sites involved, harmonising timescales for delivery and the wide-range of study materials needed in order to carry out a multinational trial. The specific focus in this paper is on study materials used in multinational trials. Study materials include all text-based documents and online content used in the study. Some but not all study materials will need to be translated into local languages.

Translating study materials can present risks to quality.[5] Cross-cultural validity may be compromised if constructs with a particular meaning in one culture are simply translated into the equivalent word in a different language. For example, the valorisation of personal empowerment is higher in individualistic than collectivist cultures,[6] so a translated standardised measure of empowerment may be subject to differing social desirability biases when used in different settings. Cultural validity can also be influenced by dialect differences. For example, words such as strike and entrée have different meanings in British English and American English, which may have relevance to research about domestic violence and food security, respectively. The emergent solution to this issue is to prioritise conceptual equivalence,[7] also termed semantic equivalence[8] or symmetrical translation,[9] defined as prioritising retention of meaning over direct transliteration. Prioritising conceptual equivalence ensures that idiomatic, cultural, and experiential equivalence is considered. The importance of ensuring linguistic consensus, defined as the process of assessing and confirming the conceptual equivalence and content validity of translations of measures,[10] is now established.

Standardised measures are a type of study material which pose a particular translation challenge. They need to be translated, cross-culturally adapted, and validated, and each step may compromise their psychometric properties.[11] This challenge is particularly relevant to global health trials, for several reasons. First, multiple measures without existing local validated translations may be needed. This is especially the case in relation to multilingual countries such as India, which has 23 official languages. Second, the cultural gap between the country in which the measure was developed and/or validated and in which it is to be used may be large, resulting in semantic, idiomatic and experiential differences.[12] Third, local research teams in any specific low-income, middle-income or high-income site may not have substantial experience of psychometric studies,[13] so training and cost implications need to be considered. Finally, our experience from developing multilanguage measures[14 15] for use in the later stage of the same study[16] suggests that the

time and human resources allocated for these processes which the funder would find acceptable is typically very limited. When the translation process is the first step in a larger study, such as preparation for a multinational randomised controlled trial, the time pressure to finalise the measures in order to be able to start the trial means that formal psychometric evaluation for each translated measure is often not feasible.

Methodologies have been developed for translation and validation of standardised measures. Two widely used guidelines come from the International Society for Pharmacoeconomics and Outcomes Research (ISPOR)[17] and Sousa and Rojjanasrirat.[9] These two guidelines were chosen due to their widespread use and complementary strengths. The ISPOR guidelines were developed through literature review and expert consultation, and with a particular focus on measures used in pharmaceutical studies. A strength is the process for harmonisation of different versions of a measure for use in multi-national studies. However, the guidelines focus solely on translation of standardised measures, so limitations include the use of an invariant methodology which does not take account of differing translation needs for different study materials, and the absence of any formal psychometric evaluation step. By contrast, the guidelines from Sousa and Rojjanasrirat do include formal psychometric evaluation, but do not describe the development process for the guidelines and do not take account of the possibility of translations into more than one language. While both guidelines propose approaches to establishing cultural validity in measure translation, they have two shortcomings. First, their integration and refinement for use in multinational studies requiring translation of both standardised measures and other types of study material across a range of languages is needed. Second, neither guideline addresses the need for proportionate psychometric evaluation of a translated, and therefore already-standardised, measure. The ISPOR guidelines do not recommend any actual use of the translated measures to test psychometric adequacy, whereas the guidelines from Sousa and Rojjanasrirat recommend involving 300–500 participants per item for full psychometric testing, a quality threshold which may hinder global health trials particularly in lower resource settings. An approach between these extremes is needed.

A proportionate approach to translation provides a solution to these challenges. By 'proportionate' we mean an approach which is based on established guidelines for translation, and involves specific steps to ensure that all study materials are translated in a way that maintains adequate quality and cultural validity, but also is sufficiently feasible in time and human resources to be used within the context of a global health trial.

A proportionate translation approach addresses two neglected issues. First, different levels of translation rigour are needed for different types of study materials. The trade-off between rigour and pragmatism is now recognised in implementation research,[18 19] but has

not yet been incorporated into translation guidelines. Second, complete psychometric evaluation of a translation of an established measure may be unnecessary when the psychometric adequacy of the measure has already been established, because the resource costs involved in conducting a complete psychometric evaluation may outweigh the scientific benefits. However, appropriate and proportionate evaluation of translated versions of standardised measures, especially of the primary outcome measure, is needed. The aim of this study was to develop a proportionate methodology for translating all types of study materials in global health trials.

## METHODS
This study took place as part of Using Peer Support in Developing Empowering Mental Health Services (UPSIDES),[20 21] a 5-year (2018–2022) European Union funded multinational trial addressing the global priority of mental health[22] by replicating and scaling-up mental health peer support interventions in order to improve social inclusion. Peer support involves people with lived experience of mental health conditions supporting others in their recovery journeys,[23] which may involve modification in different settings for global implementation.[24] Peer support is an evidence-based intervention with 19 published randomised controlled trials[25] from USA (n=12), England (n=3), Australia (n=1), Canada (n=1), Germany (n=1) and Japan (n=1). UPSIDES is divided into a preparation phase (2018–2019) including the work reported here, followed by a randomised controlled trial (2020–2022). The Coordinating Centre is Ulm, Germany.

### Study setting and sites
Mental health services in Ulm and Hamburg in Germany (local language: German; high resource setting), Kampala in Uganda (local language: Luganda; low resource setting), Dar es Salaam in Tanzania (local language: Kiswahili; low resource setting at the time of data collection, rebanded in 2020 to lower-middle resource setting), Be'er Sheva in Israel (local language: Hebrew; high resource setting) and Ahmedabad in India (local language: Gujarati; lower-middle resource setting).

### Patient and public involvement
Patients and the public including clinicians and policy-makers were involved in the design of this research. Applicants and work package leads for UPSIDES include people with lived experience of mental health issues. Patients and the public are involved in study leadership through an international advisory board and local advisory boards in each site.

### MEASURES
Although several measures were used in the UPSIDES Trial,[21] in this methodological paper we focus on the evaluation relating to the primary outcome measure which is The Social Inclusion Scale (SIS). The SIS is a 16-item service user (SU)-rated measure of social inclusion.[26] Each item is rated on a 4-point Likert scale from 1 (not at all) to 4 (yes definitely). The total score is the sum of all items, ranging from 16 (low social inclusion) to 64. The internal consistency of the English-language version was 0.85, and was acceptable for subscales of social isolation (0.76), social acceptance (0.76) and social relations (0.70). Convergent validity with other scales was adequate, ranging from 0.58 to 0.65.

### PROCEDURES
The UPSIDES study language is English, meaning that the source language for all study materials other than locally collected data is English, analysis of qualitative data is in English, and all employed UPSIDES researchers are bilingual in the local language and English. The translation team (n=4) comprised the translation work package leads and researchers from India (JK and PK) and UK (JK and MS). This team coordinated the translation tasks conducted across all sites in the preparation phase, including those relating to ethical approval processes, interviews, focus groups, the intervention manual and trial measures. All prospective participants were provided with an information sheet to read and were given the opportunity to ask questions before providing written informed consent prior to participation.

The UPSIDES Proportionate Translation Methodology was developed in three stages. In summary, Stage 1 involved the development of a typology of categories of study materials, so that the required level of rigour in translation can be identified for each study material. Stage 2 involved the development of the preliminary methodology, through integration of two established translation frameworks and expert consultation across the multinational UPSIDES research team. Stage 3 involved the implementation of the preliminary methodology within the UPSIDES study, leading to refinement to produce the final methodology. All three stages were led and coordinated by the UPSIDES translation team (n=4).

In stage 1 (Study material categories), a proposal for categorising study materials in relation to required translation rigour was presented by the translation team to UPSIDES researchers at a study meeting held in Kampala in March 2018. Participants (n=27) at the meeting came from Germany (n=6), UK (n=3), Uganda (n=11), Tanzania (n=2), Israel (n=3) and India (n=2). An initial proposal was presented for four categories of study material: training manual/online resources; study materials used across sites; qualitative data collected; and standardised measures. This was refined through discussion with study meeting participants.

In stage 2 (Methodology development), two existing translation frameworks were integrated by the translation team in order to develop a comprehensive set of translation processes. The ISPOR framework identifies ten steps for translation: preparation; forward translation;

reconciliation; back translation; back translation review; harmonisation; cognitive debriefing; review of cognitive debriefing results and finalisation; proof reading; and the final report.[17] The Sousa and Rojjanasrirat guidelines identify seven steps: forward translation; comparison of the two translated versions of the instrument; blind back-translation; comparison of the two back-translated versions; pilot testing of the prefinal version of the instrument; preliminary psychometric testing; and full psychometric testing.[9] The integration process was led by the translation team. This involved merging, discussion and iterative refinement of the two frameworks to ensure the UPSIDES Proportionate Translation Methodology was informed by best practice. Terms for similar procedures was unified and overlapping processes such as forward translation 'reconciliation' and 'comparison' were merged. The integrated translation methodology and proposals for the rigour needed for each category of study material identified in stage 1 were then discussed by UPSIDES researchers at a study meeting held in Tanzania in February 2019. Participants (n=27) came from Germany (n=6), UK (n=6), Uganda (n=2), Tanzania (n=7), Israel (n=4) and India (n=2). No formal criteria for consensus were used. There were no points of disagreement within the participants, but if any had arisen they would have been resolved through discussion within the translation team. Through this process, consensus on the preliminary UPSIDES Proportionate Translation Methodology was developed.

In stage 3 (Implementation and refinement), the preliminary UPSIDES Proportionate Translation Methodology was implemented in the five UPSIDES sites (Germany, Uganda, Tanzania, Israel, India). Each site established a local Expert Panel, with inclusion criteria for membership being: fluent speaker of English language and the local language; knowledge of Anglophone culture; familiar with terminology used in the primary outcome measure (SIS); either a mental health SU, informal carer or professional or an UPSIDES researcher. The role of the Expert panel at each site is to maximise conceptual and cultural equivalence of the SIS. To illustrate the results from our proportionate psychometric evaluation approach, we report the psychometric findings for the UPSIDES Trial primary outcome measure (SIS).[21] The final UPSIDES Proportionate Translation Methodology was refined by the translation team, based on implementation findings.

## ANALYSIS

Normality was assessed by examining histograms of the distributions of the outcomes. To assess content validity, English language and local language SIS total scores were compared using a paired t-test and Cronbach's alpha. Cronbach's alpha was used to determine if the translated versions and English version were similar in terms of range and variability across all items, and paired t-tests were used to compare the sample means of the translated and English questionnaire to test similarities in statistics

of the whole construct not content validity. To determine whether the internal consistency of the English language version was maintained, Cronbach's alpha was calculated on pooled English language data. To assess internal consistency of translations, Cronbach's alpha was calculated for local language SIS data for each language that had 10 or more ratings. An alpha above 0.6 was deemed adequate.[27] Item-level floor and ceiling effects for each local language were investigated by calculating items (ranging from 1 to 4) with highest or lowest rating endorsed by an arbitrary threshold of 75% or more participants. To assess convergent validity, a Spearman correlation matrix for all English language items was compared with the Spearman correlation matrix for all local items, and differences reported.

## RESULTS

### Stage 1 (study material categories)

The need for a proportionate approach with standardised measures was identified by participants in the study meeting. The final four agreed categories for study materials are shown in table 1.

These categories indicate an ascending importance of translation rigour, with category 0 materials (local text) requiring no translation, through categories 1 (study-generated text) and 2 (non-primary measures) to category 3 (primary outcome measure) requiring the highest quality of translation.

### Stage 2 (methodology development)

The preliminary UPSIDES Proportionate Translation Methodology and the proportionate approach for each category of study material are shown in table 2.

### Stage 3 (implementation and refinement)

Implementation of the preliminary UPSIDES Translation Methodology within the UPSIDES Study is described in table 3.

In step 1, translations of one of the secondary measure (Euroqol-5D)[28] were already available in many of the study languages. No other standardised measure had existing translations in more than one study language. Expert Panel membership comprised: Germany (n=5): researchers (n=4), administrator/translator (n=1); Uganda (n=7): psychiatrist (n=3), psychologist (n=1), peer support workers (n=2), social worker (n=1); Tanzania (n=6): researchers (n=2), clinicians (n=3), mental health SU (n=1); Israel (n=5): rehabilitation instructors (n=3), director (n=1), social work student (n=1); and India (n=5): psychiatrist (n=1), social worker (n=1), medical officer (n=1), nurse (n=1), attendant (n=2). Each site identified two bilingual members to conduct forward translations and one independent bilingual speaker with the same inclusion criteria as the Panel.

Step 2 involved forward translation of category 1a, 1b, 1c, 2 and three study materials (as described in table 1) from English to the local language. In step 3, each site convened a meeting of their Expert Panel to compare

**Table 1** Categories of study material

| Category | Description | Examples from UPSIDES study | Source language | Target language |
|---|---|---|---|---|
| 0 | Local text | Local newsletter, local language website content | Local | None |
| 1 | Study-generated text | Materials generated within the UPSIDES study | | |
| 1(a) | Research materials | Training manual (written and online content), participant information sheets, consent forms, interview or focus group topic guides, study newsletters | English | Local |
| 1(b) | Qualitative data | Interview transcripts, focus group transcripts, qualitative data from process evaluation, field notes | Local | English |
| 1(c) | Unstandardised measures | Unstandardised process/economic evaluation/ outcome measures | English | Local |
| 2 | Non-primary measures | Standardised process/economic evaluation/secondary outcome measures | English | Local |
| 3 | Primary outcome measure | Standardised primary outcome measure | English | Local |

UPSIDES, Using Peer Support in Developing Empowering Mental Health Services.

and integrate the two local language forward translations of the SIS. The Israel site chose to circulate the two local language forward translations to each Expert Panel member before the meeting and the India site circulated the finalised version to Expert Panel members after the meeting for further refinements. These approaches led to the refinement to step 3 to consult with Expert Panel members before and after the meeting.

In step 4, independent back translations of category 2 and 3 material from the local language to English were made. In step 5, the translation team compared back translations with the original and highlighted differences to be addressed by each site. In step 6, the translation team convened a single harmonisation meeting about the category 3 measure (SIS) via an online teleconference with local leads from each site.

In step 7, category 2 (n=10) and 3 (n=1) measures were pre-tested in the local language version with a target convenience sample of five SUs and five mental health workers (MHWs) fluent in the local language. Sample size achieved for each site: Germany 7 SUs, 7 MHWs; Uganda 5 SUs, 5 MHWs; Tanzania 5 SUs, 5 MHWs; Israel 5 SUs, 5 MHWs; India 5 SUs, 5 MHWs. It was made explicit that no changes could be made to the order of scales, deletion of items or questions. One site chose to carry out pretesting at two locations due to context and dialect differences in the local language. This led to refinement to step 7, to identify if more than one site is participating from the same country, in which case to discuss if local language dialect differences are important to capture and potentially to carry out pretesting at each site. This step was implemented to test the cultural and conceptual

**Table 2** Preliminary UPSIDES proportionate translation methodology

| Translation process | Source 1=Sousa and Rojjanasrirat 2=ISPOR guidelines | Category 0 Local text | Category 1 Study-generated text | Category 2 non-primary measures | Category 3 primary outcome measure |
|---|---|---|---|---|---|
| Step 1: preparation | 2 | No | Yes | Yes | Yes |
| Step 2: forward translation | 1 2 | No | Yes | Yes | Yes |
| Step 3: reconciliation | 1 2 | No | No | No | Yes |
| Step 4: back translation | 1 2 | No | No | Yes | Yes |
| Step 5: back translation review | 1 2 | No | No | Yes | Yes |
| Step 6: harmonisation | 2 | No | No | No | Yes |
| Step 7: pretesting | 1 2 | No | No | Yes | Yes |
| Step 8: finalisation | 2 | No | No | Yes | Yes |
| Step 9: psychometric evaluation | 1 | No | No | No | Yes |

ISPOR, International Society for Pharmacoeconomics and Outcomes Research; UPSIDES, Using Peer Support in Developing Empowering Mental Health Services.

**Table 3** Implementation of preliminary UPSIDES proportionate translation methodology in UPSIDES study

| Step | Implementation in UPSIDES study |
| --- | --- |
| 1.Preparation | Tasks conducted by the translation team and coordinating centre<br>Obtained permission to use each measure. Identified existing translations which do not need to be retranslated. Prepared the measures and materials needed for the trial.<br>Task conducted in each site (n=5)<br>Nominated a local translation lead to liaise with translation team. Established a local Expert Panel (comprising 5–10 members). Identified two bilingual members (B1 and B2) to conduct forward translations. Recruited independent bilingual speaker B3 with same inclusion criteria as local expert panel. Created site-specific audit files identifying) each step to be conducted. |
| 2.Forward translation | Tasks conducted in each site<br>Forward translated Category 1, 2 and 3 study materials from English to the local language by B1. Forward translated Category 3 (primary outcome measure: SIS) materials from English to the local language independently by B2. B1 and B2 sent all forward translations to the coordinating team. |
| 3.Reconciliation | Tasks conducted in each site<br>Convened a meeting of their local expert panel to compare and integrate the two local language forward translations and the English version of SIS. Identified errors, discussed discrepancies, and resolved any inadequate expressions or concepts or divergent interpretations of ambiguous terms in the English language version of SIS. Agreed on a final single consolidated forward translated version of SIS. |
| 4.Back translation | Tasks conducted in each site<br>Independent back-translation of category 2 from the local language to English by B1 or B2. Independent back-translation of category 3 material (the consolidated version of SIS) from the local language to English by B3. B1 and B3 sent all back translated versions to the coordinating team. |
| 5.Back translation review | Tasks conducted by the translation team<br>Compared and reviewed the back translation with the original English language version of all category 2 and 3 materials, focusing on cultural and conceptual equivalence rather than linguistic equivalence. Highlighted any discrepancies or problematic items that needed to be addressed or reviewed by the local translation lead.<br>Tasks conducted in each site<br>Reviewed any highlighted items, language and concepts and resolved these themselves, with their internal team or their local expert panel. Continuous liaison and communication with the coordinating team in order to iteratively refine and agree the final version of the materials, including finalisation of category 2 measures for pretesting and the category 3 measure (SIS) for harmonisation. |
| 6.Harmonisation | Preparation for the harmonisation meeting by the translation team<br>Convened a single harmonisation meeting about the category 3 measure (SIS) via an online teleconference with all local translation leads representing each language.<br>Content of the harmonisation meeting<br>Each site provided verbal back translation on the different components of the local language version of the SIS including the instructions, each individual item, and response format. Jointly identified and addressed any translation discrepancies or conceptually problematic items that arose between different local language versions of SIS, to maximise conceptual and cultural equivalence between the English and local language versions. Components were identified that required further local discussion, for example, with the local UPSIDES team or local expert panel.<br>Tasks conducted after the harmonisation meeting by each site<br>Local discussion, where needed, was held. Final changes were sent to the coordinating team. |
| 7.Pretesting | Tasks conducted by each site<br>Category 2 and 3 measures were pre-tested in the local language version with a target convenience sample of 5 service users fluent in the local language (for service user-rated measures) and five mental health workers fluent in the local language (for staff-rated measures). Participants were asked to complete the measure and rate (1) administration instructions, (2) the response format and (3) each item using a dichotomous scale (CLEAR or UNCLEAR). Semistructured interviews were then conducted by UPSIDES research workers with the participant. The topic guide explored each component rated as UNCLEAR, and views about the measure as a whole in relation to clarity, ease of completion, usability, cultural validity and translation alternatives. Numerical scores for the measure were recorded. Suggested modifications and aspects which required further review were transcribed into English and recorded on a preprepared spreadsheet. These were sent to the coordinating team.<br>Tasks conducted by the translation team<br>Review the data from each site for anomalies, including language version used and data distribution from completed measures. |

Continued

| Table 3 | Continued |
|---|---|
| **Step** | **Implementation in UPSIDES study** |
| 8. Finalisation | Tasks conducted by the translation team<br>Proposed modifications identified in pretesting were discussed with translation leads, and any resulting actions agreed. Possible actions were: implement proposed change; consult with site lead or local expert panel; do not implement proposed change. The decision about each proposed modification was recorded.<br>Tasks conducted by each site<br>Changes were highlighted in both the English and local language version of the measure to clearly indicate where changes were made. Following agreement on changes, the translation was finalised. |
| 9. Psychometric evaluation | Tasks conducted by each site<br>Psychometric testing was undertaken for category 3 (SIS). A sample of up to 20 participants per site were recruited. Inclusion criteria: aged over 18 years, currently using mental health services, fluent in local language and English, and able to give consent as judged by a clinician. Participants completed the local language SIS measure without seeing the English version and then the English-language SIS in which items had been reordered to reduce practice effects. Scores were recorded on a preprepared spreadsheet.<br>Tasks conducted by the translation team<br>Coordinated the analysis of data from all sites. |

SIS, Social Inclusion Scale; UPSIDES, Using Peer Support in Developing Empowering Mental Health Services.

equivalence of UPSIDES measures and identify any aspects that were unclear and then amend to enhance real-world applicability.

In step 8 (finalisation), 197 proposals (Germany 134, Uganda 10, Tanzania 11, Israel 14, India 28) for modification were considered, included changed wording (eg, simplifying, using gender-inclusive language), review a specific word/sentence for syntactic or semantic correctness, add or change guidance examples to improve cultural relevance, for example, add 'synagogue' or change from 'general practitioner' to 'family' or 'general doctor', amend instruction format, for example, add extra guidance 'this is about your opinions', and spelling. Overall, 155 proposals were reviewed and implemented by the translation team. Non-implemented proposals (n=42) included scale modifications and proposals better addressed by amending rater instructions.

In step 9, SIS was completed in both local language and English by 34 participants (Germany 11, Tanzania 11, Israel 7, India 5) and in English language only by 20 participants at the Uganda site since it emerged that this site only needed an English language version of SIS. This led to the refinement to step 1 to identify what language version is actually needed, rather than simply asking sites to identify their local language. The psychometric evaluation of SIS is shown in table 4.

There was no significant difference between SIS score in English and any translated version, and Cronbach's alpha comparing all English and local language SIS scores was high, indicating adequate content validity. Cronbach's alpha for all English language, pooled local languages, and the two translations with sufficient local language data to calculate, all exceeded predefined thresholds, indicating adequate internal consistency. Item-level floor and ceiling effects were minimal. Correlation matrices were very similar for English language compared with German, Kiswahili and Hebrew, indicating adequate convergent validity. The exception was Gujarati, in which

marked differences were shown in responses to one item (#3) and smaller differences for three other items (#11/12/14).

The preliminary UPSIDES Proportionate Translation Methodology was refined based on the implementation findings, including obtaining early agreement with measure developers about copyright and dissemination approaches, more focus on ensuring translation needs have been accurately identified, increased clarity and monitoring to reduce inconsistencies in data collection, and introduction of a new dissemination step. The final UPSIDES Proportionate Translation Methodology is shown in table 5.

In the final UPSIDES Proportionate Translation Methodology, steps 1–3 develop the forward translation in the local language, steps 4–6 refine through back-translation, steps 7–9 evaluate the standardised measures and step 10 involves knowledge mobilisation strategies.

## DISCUSSION

In this study, we developed a typology of study materials relating to global health trials, and then evaluated and refined a proportionate methodology for translating these study materials. UPSIDES is a large and relatively well-resourced study led by a committed team of investigators. However, the challenges of translating the various study materials to a sufficient quality were significant. The large number of stakeholders across all sites added further complexity. Proportionate approaches to translation are needed, given that the resources available for translation processes can be limited within the study and can differ between sites.

A priority-setting exercise involving 412 participants from 80 countries developed a global health trials methodological research agenda.[29] The priority most commonly rated as critically important was choosing appropriate outcomes to measure. A key insight arising

**Table 4** Psychometric evaluation of the Social Inclusion Scale (SIS)

| | All translations | German | Luganda | Kiswahili | Hebrew | Gujurati |
|---|---|---|---|---|---|---|
| **Content validity** | | | | | | |
| SIS (English) mean (SD) | 48.5 (7.5) | 51.1 (4.8) | | 50.2 (9.6) | 46.0 (3.2) | 42.8 (8.0) |
| SIS (local) mean (SD) | 49.3 (7.8) | 51.9 (5.3) | | 51.5 (10.2) | 46.7 (2.4) | 42.8 (6.8) |
| Significance (p) | 0.08 | 0.07 | | 0.27 | 0.42 | 1.00 |
| English SIS vs local SIS Cronbach's alpha (p) | 0.97 (<0.001) | 0.98 (<0.001) | | 0.96 (<0.001) | 0.88 (0.011) | 0.97 (0.001) |
| **Internal consistency** | | | | | | |
| English Cronbach's alpha (p) | 0.70 (<0.001) | | | | | |
| Local Cronbach's alpha (p) | 0.73 (<0.001) | 0.63 (0.006) | | 0.87 (<0.001) | | |
| **Item-level floor effects** | | | | | | |
| English (n) | | 0 | 0 | 0 | 0 | 1 |
| Local (n) | | 0 | 0 | 0 | 0 | 1 |
| **Item-level ceiling effects** | | | | | | |
| English (n) | | 0 | 0 | 2 | 4 | 4 |
| Local (n) | | 2 | 0 | 2 | 4 | 5 |

from our methodology is that this choice involves not only considering the scientific rationale, but also the availability of existing translations and the costs involved in translating. Although not formally measured, it was observed that the resources needed to translate longer and more conceptually complex standardised measures were markedly higher than for short simple measures, due to the multiple rounds of discussions, the difficulties of ensuring conceptual equivalence, and harmonisation challenges. Future research could develop an empirically based metric of these features which influence translation costs, to be considered alongside other features of the measure such as psychometric adequacy and translation availability, in selecting measures for use in global health trials.

A strength of the study is the development of a typology to allocate each study material to a category, ensuring a shared understanding about translation needs. For example, in UPSIDES, it was only through the application of this approach that it was confirmed that local language newsletters do not need to be translated. This reduces wasted effort. A second strength is our field-testing of the methodology. The ISPOR guidelines have recently been updated,[10] but only using a consensus process. In UPSIDES, we found that significant human resources were needed to coordinate the translation processes, and these need to be included where possible in future trials requiring translation of study materials. In less-resourced

global health studies, hard choices need to be made about which aspects of quality to emphasise,[30] and our methodology provides a framework to inform these choices.

Several limitations can be identified. First, the methodology was implemented by its developers, meaning that some components of the methodology may be implicit knowledge which is not sufficiently described. The methodology is based on refinement of two existing widely used methodologies, and other guidelines could also have been considered.[31] Second, the generalisability to global health trials in areas of medicine beyond mental health is un-known. Future research with other clinical populations might explore population-specific differences to identify the generalisability of the UPSIDES Proportionate Translation Methodology. Third, several implementation challenges occurred, such as difficulties in identifying enough bilingual SUs in some sites. We believe that reporting these challenges explicitly, as we have done, is preferable to the alternative—and perhaps more common—approach of not reporting any details about measure translation processes. Finally, we were only able to demonstrate preliminary psychometric adequacy of the primary outcome SIS due to the limited sample sizes for each site. For example, we did not attempt to assess discriminative or construct validity. It could be argued that this is indefensible, and only quality-endorsed translation of measures from an established standardised core outcome sets such as Core Outcome

**Table 5** Final version of UPSIDES proportionate translation methodology

| Step | Process | Considerations |
|---|---|---|
| Step 1: Preparation | Site tasks<br>Create a local expert panel (including bilingual members B1 and B2) and a bilingual speaker B3 at each site. B1–B3 should have the target language as their native language and be fluent in the source Language. Ensure all stakeholders are familiar with categories of study materials. Identify the most widely used language at the local site to avoid developing local language versions which will not be used<br>Translation team tasks<br>Allocate study materials to categories 1–3. Obtain permission to use materials, for example, confirming background and foreground intellectual property arrangements, copyright and other statements to be included in translations, and dissemination plan such as freely available from study web-site. Collate existing translations for measures, identifying where new translations are needed. Create comprehensive step-by-step site-specific audit files with accessible instructions, such as text, webinar or individual training. | Preparing additional learning materials needed to support sites with the complexity of translation tasks. |
| Step 2: Forward translation | Site tasks<br>Category 1–3 material is forward translated from the source language to the target language by B1. Category 3 material is independently forward translated from the source language to the target language by B2. | Ensure all parts of measure are translated, including administration instructions and rating scales. Allocate enough time. |
| Step 3: Reconciliation | Site tasks<br>Circulate the two forward local language translations to Expert Panel members with a preprepared table to examine each item, identify problematic items and offer alternative translations in preparation for the expert panel. Integrate each Expert panel member's table into one table to highlight different responses, suggestions and problematic items for which there was a lack of consensus. The more items for which different opinions arose the more time must be allocated for the Expert Panel. The Expert Panel meet to compare and integrate the two forward translation target language versions and the source language version for the category3 measure. Discussion may identify errors and discrepancies, resolve inadequate expressions or concepts, and integrate interpretations of ambiguous terms in the source language version. After consensus is reached, produce a forward translation of the category 3 primary outcome measure (POMv1). This version can be circulated to members following the meeting for further review (optional). Send minutes of the meeting to the translation team. | Speak with local stakeholders to identify potential participants. Try to ensure equal gender proportion and members who represent different groups. It is helpful if members are familiar with the measure and its use. Circulate the two forward local language translations to members before the meeting. Find a suitable time and day for all participants to attend. Consider honorarium for participants since it was challenging for some sites to recruit members. One person should chair the meeting and encourage inclusive conversation among members. If Expert Panel final version is circulated for further comments, instead of convening another meeting carry out follow-up interviews/meetings with individual Expert Panel members for comments before step 4. Depending on the number of items, enough time needs to be scheduled or more than one Expert Panel meeting arranged. |
| Step 4: Back translation | Site tasks<br>B1 back translates the category 2 measures and POMv1 into the source language. B3 independently back translates POMv1 into the source language, with a request to identify constructs that are subjective, culturally sensitive or specific that might need to be translated more conceptually. | Some sites found it difficult to find translators without payment so a study budget for translators may be needed. Allocate enough time. |
| Step 5: Back translation review | Translation team tasks<br>The translation team compare the back-translated versions of the category 2 measures and the category 3 POMv1 measure with the source language versions. Consider the similarity of instructions, items, and response format, with a focus on cultural/ conceptual equivalence (ie, meaning and relevance) rather than linguistic equivalence (ie, exact wording, sentence structure).<br>Site tasks<br>If discrepancies cannot be resolved, items that do not retain their original meaning are retranslated and back-translated. Finalise all category 2 measures and refine the category 3 POMv2. | Repeated rounds of coordination between the translation team and sites is needed. |
| Step 6: Harmonisation | Translation team tasks<br>One harmonisation meeting is convened (eg, by phone or video call) with all sites to discuss POMv2. Each site provides verbal comments on all back translations. The translation team facilitate a discussion to identify and address any translation discrepancies or conceptually problematic items that arise between different language versions, and to ensure conceptual and cultural equivalence between the source language and the target language version. Harmonise the category 3 measure across all sites to produce POMv3. | Choose a familiar platform all sites can access to attend. Coordinate with sites in advance to arrange a time. Ensure decision-making process is agreed in advance, for example, final decision is made by consensus, or by translation team |

**Table 5** Continued

| Step | Process | Considerations |
|---|---|---|
| Step 7: Pre-testing | Site tasks<br>Recruit up to five participants from the target population. After consent, ask them to complete the Category 2 measures and POMv3 measure using a think-aloud protocol, rate each component (including administration instructions, item content and rating scale) for clarity, and then record an interview about each component rated as unclear and about the measure as a whole, including cultural validity, translation alternatives, and the cognitive equivalence of the translation. Send a transcription of the proposed modification and major issues identified in the interview and field notes to the translation team. | Can be a lengthy process with lots of measures particularly with service users. May highlight measures that are not culturally appropriate, which needs a strategic (rather than translation team) decision. If more than one site is participating using the same language, then discuss whether between-site dialect differences are significant. If they are, then carry out pretesting at each site. |
| Step 8: Finalisation | Translation team tasks<br>The pre-testing results are reviewed and any further modifications are discussed with the local teams and actioned. The category 2 measures and the POMv4 are finalised. | Ensure both (A) prefinal versions with changes highlighted and (B) final versions are archived, to allow complete audit trail. |
| Step 9: Psychometric evaluation | Site tasks<br>Recruit up to 20 participants from the target population with additional inclusion criterion of being fluent in both the local language and the source language. Participants complete POMv4 without seeing the source language version, and then the source language version in which items have been reordered to reduce practice effects. Send quantitative and qualitative data from each site to the translation team.<br>Translation team tasks<br>Analyse, for example, to assess convergent validity, construct validity, internal consistency and floor and ceiling effects. Identify if psychometric performance is so poor that refinement and re-testing is needed. | Difficulties with recruiting bilingual speakers may mean sample size needs to be reduced. |
| Step 10: Dissemination | Translation team tasks<br>Prepare the final version of all study materials, including content (eg, copyright statement) agreed in Step 1, in non-editable form, for example, as a PDF. Disseminate the final version of all study materials to the study group, for example, on the private side of the study web-site. If agreed in Step 1, make measures available as downloads, for example, on the public side of the study web-site. | Proof-read to check no typographical errors are introduced. |

B1–B3=bilingual speakers of source and target language.
POMv1–POMv4=primary outcome measure version 1–4.
UPSIDES, Using Peer Support in Developing Empowering Mental Health Services.

Measures in Effectiveness Trials (COMET)[32] should be used. However, a systematic review of standardised mental disorder screening tools found that no psychometrically-established measure existed for over 100 low-income and middle-income countries.[33] Furthermore, the limited involvement of research groups from lower-resource settings in COMET[29] highlights that the creation of a multi-language core outcome set is a long-term goal. In the short term, global health trials cannot wait until rigorous translations of measures in all site languages are developed and published. Ensuring adequacy in translation quality needs to be balanced with the limited availability of time and resources. For example, comparison between the two bilingual translations and/or using an automatic translator to compare the two human translations would have added further rigour as would more detailed specification of language proficiency, subject matter knowledge and translation training.

## CONCLUSION

The UPSIDES Proportionate Translation Methodology can be used to inform the translation, cross-cultural adaptation and validation of all study materials in future multisite global trials working with a range of sites with differing resources. The categorisation framework can be used to establish the levels of translation rigour needed for different types of study material. The methodology supports a stronger emphasis on quality for the study materials, such as the primary outcome measure, which require the most rigorous translation and validation. Overall, this study contributes to the goal of ensuring optimal use of available resources in global health trials.

**Author affiliations**
[1]School of Health Sciences, Institute of Mental Health, University of Nottingham, Nottingham, UK
[2]Centre for Mental Health Law and Policy, Indian Law Society, Pune, Maharashtra, India
[3]School of Medicine, University of Nottingham, Nottingham, UK
[4]Department of Social Work, Ben Gurion University of the Negev, Be'er Sheva, Israel
[5]Department of Psychiatry II, Ulm University, Ulm, Germany
[6]Department of Psychiatry, University Medical Center Hamburg-Eppendorf, Hamburg, Germany
[7]Butabika National Referral Hospital, Kampala, Uganda
[8]Makerere University, Kampala, Uganda
[9]Department of Health Systems, Impact Evaluation and Policy, Ifakara Health Institute, Ifakara, Morogoro, United Republic of Tanzania

[10]Centre for Global Mental Health, London School of Hygiene and Tropical Medicine, London, UK

**Acknowledgements** MS acknowledges the support of the Centre for Mental Health and Substance Abuse, University of South-Eastern Norway, the NIHR Nottingham Biomedical Research Centre, and research group work support from the Economic and Social Research Council (grant numbers ES/J500100/1 and ES/P000711/1). AC is supported by the Economic and Social Research Council (grant number ES/P000711/1).

**Contributors** AC and PK contributed equally and share the first authorship. BP was the principal investigator. AC, PK, RH, CM, GM, JNa, BP, GKR, DS, JK and MS contributed to study design. CN led the statistical analyses. AC, PK, BP, JK and MS coordinated the methodology development, supervised data collection, conceptualised the manuscript, and wrote the first draft of the manuscript. AC, PK, CN, AG, RH, CM, GM, JNa, JNi, RN, EP, BP, MR, GKR, DS, JK and MS contributed to data interpretation, provided critical input and approved the final version of the manuscript. AC is the gurantor and accepts full responsibility for the work and/or the conduct of the study, had access to the data, and controlled the decision to publish.

**Funding** UPSIDES has received funding from the European Union's Horizon 2020 Research and Innovation Programme under Grant Agreement No 779 263. This publication reflects only the authors' view. The Commission is not responsible for any use that may be made of the information it contains.

**Competing interests** None declared.

**Patient consent for publication** Not required

**Ethics approval** The study was approved by the ethics committee in each of the respective countries including Ulm University Ethics Commission (Application nr. 195/18), Ärztekammer Hamburg, Germany (MC-230/18), Mengo IRB Uganda (MH: 360; MH/REC/141/8/2018), National Institute for Medical Research Tanzania (NIMR/HQ/R.8a/Vol.IX/2982), Institutional Review Board, Ifakara Health Institute, Tanzania (IHI/IRB/No. 28-2018), Human Subjects Research Committee of Ben-Gurion University (ref: 1621-2), Indian Council of Medical Research (Indo-foreign/66 /M/2017-NCD-1) and Indian Law Society (ILS/37/2018).

**Provenance and peer review** Not commissioned; externally peer reviewed.

**Data availability statement** The data that support the findings of this study are available in the repository OPARU at: https://dx.doi.org/10.18725/OPARU-40623.

**ORCID iDs**
Ashleigh Charles http://orcid.org/0000-0003-2222-4358
Chris Newby http://orcid.org/0000-0002-2936-8592
Ramona Hiltensperger http://orcid.org/0000-0003-2544-4188
Candelaria Mahlke http://orcid.org/0000-0001-9573-6106
Galia Moran http://orcid.org/0000-0001-9718-1773
Juliet Nakku http://orcid.org/0000-0002-0611-1102
Rebecca Nixdorf http://orcid.org/0000-0002-3064-8380
Eva Paul http://orcid.org/0000-0002-5827-0578
Bernd Puschner http://orcid.org/0000-0002-2929-4271
Grace Kathryn Ryan http://orcid.org/0000-0002-9310-3513
Donat Shamba http://orcid.org/0000-0001-7431-7199
Jasmine Kalha http://orcid.org/0000-0001-7357-2366
Mike Slade http://orcid.org/0000-0001-7020-3434

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
