## [Reviewer comments · BMJ Open]

ARTICLE DETAILS

TITLE (PROVISIONAL)	Proportionate translation of study materials and measures in a multinational global health trial: methodology development and implementation
AUTHORS	Charles, Ashleigh; Korde, Palak; Newby, Chris; Grayzman, Alina; Hiltensperger, Ramona; Mahlke, Candelaria; Moran, Galia; Nakku, Juliet; Niwemuhwezi, Jackie; Nixdorf, Rebecca; Paul, Eva; Puschner, Bernd; Ramesh, Mary; Ryan, Grace; Shamba, Donat; Kalha, Jasmine; Slade, Mike

VERSION 1 – REVIEW

REVIEWER	Wiangkham, Taweewat University of Birmingham, Department of Physical Therapy
REVIEW RETURNED	14-Jul-2021

GENERAL COMMENTS	Thank you for the opportunity to review this manuscript. Overall, this study had collaborated with several countries. However, I feel that the manuscript has several problems in terms of unclear writing, provided inadequate information and methodological weaknesses (e.g. study design and analysis). These can lead to reduce confidence in findings. Introduction  • Paragraph 1: I am not clear what is your key message to communicate to readers? • Page 5, line 10: could you provide a little bit more about why multinational co-operation in clinical trials is essential? • I feel paragraph 2 can be improved to help readers? Furthermore, authors have mentioned semantic and conceptual equivalences. How about idiomatic and experiential equivalences? • I feel paragraph 3 can be improved to help readers? Furthermore, I think that guidelines for the process of cross-cultural adaptation of self-report measures by Beaton et al. is a one of widely common use. Why do authors consider only 2 guidelines? • I feel that your introduction cannot convince me. It may be come from inadequate information, insufficient strong rationales and missing some key evidence? Methods  • Measures  o Do you have only 2 outcome measures? o Is it possible to provide validity and reliability of the SIS? o For the EQ-5D, author used 5 levels or 3 levels? I think that authors can provide a short information. • Procedures  o How to get participant consent?
--

	o How to obtain the consensus? Any criteria?  • Analysis o Normality test? o Why was content validity assessed by a paired t-test and Cronbach's alpha? o How about a reference of your criteria for floor and ceiling effects? o How about discriminative and construct validity? o Other analyses to answer your research question/enhance your confidence in findings? Results  • Page 8, line 8: Authors have mentioned expert panel membership. Did you have criteria for the experts? • I feel that this section is difficult to follow. Where do 1a, 1b, 1c, 2 and 3 come from? Discussion  • Inadequate discuss in key points?
--	--

REVIEWER	Nakhostin Ansari, Nouredin School of Rehabilitation, Tehran University of Medical Sciences, Physiotherapy
REVIEW RETURNED	16-Jul-2021

GENERAL COMMENTS	This manuscript is aimed to develop a proportionate methodology for the translation, cross-cultural adaption, and validation of study materials in global health trials. The authors concluded the methodology can be recommended for use in multi-national investigations. This is an important and interesting manuscript describing the methods in detail and fully acknowledging the limitations.
---

REVIEWER	Dzulkarnain, A A A International Islamic University Malaysia, Audiology and Speech- Language Pathology
REVIEW RETURNED	21-Sep-2021

GENERAL COMMENTS	Thank you for the chance to review this manuscript. I am commenting this paper not as someone who expert in Psychiatry or mental health but rather someone who has some experience in doing Psychometric study, translation and cultural adaptation of the questionnaire. I would say this paper has some merit but certain part of the paper should be written in clear manner to improve its clarity (especially for those who has non-Psychiatry background). Among my general comment: Introduction/literature review  -A clear explanation on proportionate translation and its important is needed. - From my understanding, the authors argue that the psychometric property is not necessary for a translated scale since it came from well-established scale in the original language. Somehow if my point is correct, i do think the authors did not explicitly well explained this in the introduction. Methodology  -To be honest it is hard to understand the content of the 9 stages mentioned by the authors. It is well written in English but somehow it
---

	has lack of clarity -I suggest for the authors to share this paper with someone outside of their field of expertise at least to improve the clarity of the methodology. The explanation somehow is only suitable for those who know the research very well but not a wide audience of the journal. -Please rewrite to ensure the study can be repeated. -Use of flowchart maybe needed here to explain the stages Discussion -Since the methods and results were not clear, it is hard to follow this section
--	---

VERSION 1 – AUTHOR RESPONSE

Reviewer 1

- INTRODUCTION Paragraph 1: I am not clear what is your key message to communicate to readers?

Amended as suggested (Introduction, paragraph 1, 'The translation of...').

- Page 5, line 10: could you provide a little bit more about why multi-national co-operation in clinical trials is essential?

Amended as suggested (Introduction, paragraph 2, 'Close co-ordination is needed...').

- I feel paragraph 2 can be improved to help readers?

Amended as suggested by re-structuring into two paragraphs and more clearly presenting the argument (Introduction, paragraphs 2 and 3).

- Furthermore, authors have mentioned semantic and conceptual equivalences. How about idiomatic and experiential equivalences?

Added as suggested (Introduction, paragraph 3, 'Prioritising conceptual...').

- I feel paragraph 3 can be improved to help readers?

As suggested, we have edited the paragraph throughout to improve clarity.

- Furthermore, I think that guidelines for the process of cross-cultural adaptation of self-report measures by Beaton et al. is a one of widely common use. Why do authors consider only 2 guidelines?

We now clarify why these two guidelines were chosen (Introduction, paragraph 5, 'These two guidelines were chosen...') and have noted the potential use of the Beaton guidelines as a limitation (Discussion, Limitations section, 'The methodology is...').

- I feel that your introduction cannot convince me. It may be come from inadequate information, insufficient strong rationales and missing some key evidence?

We now summarise and strengthen the rationale (Introduction, final two paragraphs).

- METHODS Do you have only 2 outcome measures?

We now clarify the reason for focussing on SIS (Methods, Measures section).

- Is it possible to provide validity and reliability of the SIS?

Added as suggested (Methods, Measures section).

- For the EQ-5D, author used 5 levels or 3 levels? I think that authors can provide a short information.

Now deleted from Methods, Measures section.

- How to get participant consent?

Added as suggested (Methods, Procedures paragraph 1).

- How to obtain the consensus? Any criteria?
Added as suggested (Methods, Procedures paragraph 4).
- Analysis: Normality test?
Added as requested (Methods, Analysis section).
- Why was content validity assessed by a paired t-test and Cronbach's alpha?
Clarified as requested (Methods, Analysis section).
- How about a reference of your criteria for floor and ceiling effects?
Clarified as requested (Methods, Analysis section).
- How about discriminative and construct validity?
We now note these two properties as examples of psychometric properties which are not investigated using our proportionate methodology (Discussion, Limitations section, 'For example...').
- Other analyses to answer your research question/enhance your confidence in findings?
We now note the need for future research to establish generalisability of the methodology (Discussion, Limitation section, 'Future research...').
- RESULTS Page 8, line 8: Authors have mentioned expert panel membership. Did you have criteria for the experts?
Added as suggested (Methods, Stage 3, 'Each site ...').
- I feel that this section is difficult to follow. Where do 1a, 1b, 1c, 2 and 3 come from?
We now summarise the study material categories to make them more visible (Results, Stage 1) and we refer to Table 1 when these categories are used (Results, Stage 3, '(as described in...').
- DISCUSSION Inadequate discuss in key points?
We have now highlighted and amplified the two key points (Discussion, paragraph 2, 'A key insight...'; Discussion, paragraph 3, 'and these need...').

Reviewer 2

- This manuscript is aimed to develop a proportionate methodology for the translation, cross-cultural adaption, and validation of study materials in global health trials. The authors concluded the methodology can be recommended for use in multi-national investigations. This is an important and interesting manuscript describing the methods in detail and fully acknowledging the limitations.

We thank the reviewer for their comments.

Reviewer 3

- I would say this paper has some merit but certain part of the paper should be written in clear manner to improve its clarity (especially for those who has non-Psychiatry background). We have revised the whole paper to improve clarity. For example, we have amplified the key messages in the Conclusion (Discussion, final paragraph).
- INTRODUCTION/LITERATURE REVIEW A clear explanation on proportionate translation and its important is needed.
Added as suggested (Introduction, last two paragraphs).
- B) From my understanding, the authors argue that the psychometric property is not necessary for a translated scale since it came from well-established scale in the original language. Somehow if my point is correct, i do think the authors did not explicitly well explained this in the introduction. The reviewer is correct and we now clarify this aspect more explicitly (Introduction, last two paragraphs).
- METHODOLOGY To be honest it is hard to understand the content of the 9 stages mentioned by the authors. It is well written in English but somehow it has a lack of clarity - I suggest for the

authors to share this paper with someone outside of their field of expertise at least to improve the clarity of the methodology. The explanation somehow is only suitable for those who know the research very well but not a wide audience of the journal.

We have addressed this important point by summarising the aim of each group of steps (Results, last paragraph).

- Please rewrite to ensure the study can be repeated.

We have made multiple clarifications in the Methods section to enable the study to be replicated.

- Use of flowchart maybe needed here to explain the stages

To improve clarity we have added a text-based overview of the stages (Methods, Procedures paragraph 2).

- DISCUSSION Since the methods and results were not clear, it is hard to follow this section

We hope that the clarifications we have added throughout the paper, including an amplified Conclusions section, make the paper easier to follow.

VERSION 2 – REVIEW

REVIEWER	Dzulkarnain, A A A International Islamic University Malaysia, Audiology and Speech-Language Pathology
REVIEW RETURNED	20-Oct-2021

GENERAL COMMENTS	Thank you for the chance to re-review the paper. The authors have addressed all concerns in the previous review. Thank you.
---

REVIEWER	Daniele, Franca “G. d’Annunzio” University
REVIEW RETURNED	27-Oct-2021

GENERAL COMMENTS	The work is extremely interesting, as it attempts to introduce a new translation method that could be halfway between the ISPOR guidelines and those from Sousa and Rojjanasrirat. Eventually, the article is about translating language, so you should better explain and provide examples of the language used for the measures and study material you apply to your investigation. In the measures section - you should report the Social Inclusion Scale, as it would be easier for readers to understand it. The translators have been totally neglected; while their characteristics should be reported (e.g. besides their level of English also their knowledge of the subject matter, and their specific training), as these traits are crucial to successful translations. It is well accepted that translators should translate into their native language, should have a good knowledge of the subject matter, and should be fluent in the source language. In my opinion, levels B1 and B2 are not sufficient to translate such domain-specific material. Since these are the translators available for the investigation, I suggest that you make a comparison between the translations provided by level B1 translators and those provided by level B2 translators. Another easy translation could be undertaken using automatic translators. The translations would yield a further set of data that could be compared with those received from human
---

	translators. Consistently, also the coordinating team and experts should be better identified in the paper. I am afraid that the elevated number (which is indeed exceedingly high) of people involved not only does not reduce costs but it might create confusion in both the analytical and the executive phases of the translation method described. What do you mean by cultural and conceptual equivalence? What were the methods used to test and apply them? If this methodology is meant to be widely accepted and used, it seems to me that the approach is too subjective and it could vary with varying translation material, translators, teams, and experts, just like in all translation processes! I appreciate the effort, but what is the advantage? On the contrary, the CLEAR - UNCLEAR feedback seems especially useful, because it is a way to test the results and bring them into reality. This is not very common in translation processes. Finally, the references should comprise bibliographical citations from the areas of translation studies, in order to better support your translation choices.
--	--

VERSION 2 – AUTHOR RESPONSE

Reviewer 1

- Thank you for the chance to re-review the paper. The authors have addressed all concerns in the previous review. Thank you.

We thank the reviewer for their comments in the previous review.

Reviewer 2

- The work is extremely interesting, as it attempts to introduce a new translation method that could be halfway between the ISPOR guidelines and those from Sousa and Rojjanasirat.

We thank the reviewer for their comments.

- Eventually, the article is about translating language, so you should better explain and provide examples of the language used for the measures and study material you apply to your investigation. We have added further examples of the language used for the measures and study material (Results, paragraph 9).

- In the measures section - you should report the Social Inclusion Scale, as it would be easier for readers to understand it.

We have reported the Social Inclusion Scale in the measures section as suggested (Measures, paragraph 1).

- The translators have been totally neglected; while their characteristics should be reported (e.g. besides their level of English also their knowledge of the subject matter, and their specific training), as these traits are crucial to successful translations. It is well accepted that translators should translate into their native language, should have a good knowledge of the subject matter, and should be fluent in the source language. In my opinion, levels B1 and B2 are not sufficient to translate such domain-specific material. Since these are the translators available for the investigation, I suggest that you make a comparison between the translations provided by level B1 translators and those provided by level B2 translators. Another easy translation could be undertaken using automatic translators. The translations would yield a further set of data that could be compared with those received from human translators.

We thank the reviewer for their comments and have added these as a limitation of the study (Discussion, last paragraph).

- Consistently, also the coordinating team and experts should be better identified in the paper. I am afraid that the elevated number (which is indeed exceedingly high) of people involved not only does not reduce costs but it might create confusion in both the analytical and the executive phases of the translation method described.

We have now better identified the translation team in the paper by stating the number of members (Procedures, paragraph one, paragraph two, and paragraph four).

We have now better identified the role of the Expert Panel (Procedures, paragraph five).

We acknowledge the issue of elevated number of people involved (Discussion, paragraph one).

- What do you mean by cultural and conceptual equivalence? What were the methods used to test and apply them? If this methodology is meant to be widely accepted and used, it seems to me that the approach is too subjective and it could vary with varying translation material, translators, teams, and experts, just like in all translation processes! I appreciate the effort, but what is the advantage? We now define conceptual equivalence and locate cultural equivalence (Introduction, paragraph three).

We have now highlighted that the method used to test and apply these and amplified the advantages of this (Results, paragraph eight).

- On the contrary, the CLEAR - UNCLEAR feedback seems especially useful, because it is a way to test the results and bring them into reality. This is not very common in translation processes.

We thank the reviewer for their comments.

- Finally, the references should comprise bibliographical citations from the areas of translation studies, in order to better support your translation choices.

We now clarify that the two established frameworks informed our methodology development (Procedures, paragraph four).

VERSION 3 – REVIEW

REVIEWER	Daniele, Franca "G. d'Annunzio" University
REVIEW RETURNED	21-Nov-2021
GENERAL COMMENTS	Thank you for the opportunity to re-review the article. Although you have decided to not compare the two translators as well as with online translators, the fact that you specified it as a limitation is acceptable. It would be interesting though to see the results of such comparisons, maybe in a future study, containing more references relevant to translation studies.